# Effectiveness of COVID-19 Vaccines over 13 Months Covering the Period of the Emergence of the Omicron Variant in the Swedish Population

**DOI:** 10.3390/vaccines10122074

**Published:** 2022-12-05

**Authors:** Yiyi Xu, Huiqi Li, Brian Kirui, Ailiana Santosa, Magnus Gisslén, Susannah Leach, Björn Wettermark, Lowie E. G. W. Vanfleteren, Fredrik Nyberg

**Affiliations:** 1School of Public Health and Community Medicine, Institute of Medicine, Sahlgrenska Academy, University of Gothenburg, 405 30 Gothenburg, Sweden; 2Department of Infectious Diseases, Institute of Biomedicine, Sahlgrenska Academy, University of Gothenburg, 405 30 Gothenburg, Sweden; 3Region Västra Götaland, Department of Infectious Diseases, Sahlgrenska University Hospital, 413 45 Gothenburg, Sweden; 4Department of Microbiology and Immunology, Institute of Biomedicine, Sahlgrenska Academy, University of Gothenburg, 405 30 Gothenburg, Sweden; 5Department of Clinical Pharmacology, Sahlgrenska University Hospital, 413 45 Gothenburg, Sweden; 6Pharmacoepidemiology & Social Pharmacy, Department of Pharmacy, Uppsala University, 752 36 Uppsala, Sweden; 7COPD Center, Department of Respiratory Medicine and Allergology, Sahlgrenska University Hospital, 413 45 Gothenburg, Sweden; 8Department of Internal Medicine and Clinical Nutrition, Institute of Medicine, Sahlgrenska Academy, University of Gothenburg, 405 30 Gothenburg, Sweden

**Keywords:** SARS-CoV-2, COVID-19 vaccines, vaccine effectiveness, Omicron

## Abstract

Background: We estimated real-world vaccine effectiveness (VE) against COVID-19 infection, hospitalization, ICU admission, and death up to 13 months after vaccination. VE before and after the emergence of Omicron was investigated. Methods: We used registered data from the entire Swedish population above age 12 (*n* = 9,153,456). Cox regression with time-varying exposure was used to estimate weekly/monthly VE against COVID-19 outcomes from 27 December 2020 to 31 January 2022. The analyses were stratified by age, sex, and vaccine type (BNT162b2, mRNA-1273, and AZD1222). Results: Two vaccine doses offered good long-lasting protection against infection before Omicron (VE were above 85% for all time intervals) but limited protection against Omicron infection (dropped to 43% by week four and no protection by week 14). For severe COVID-19 outcomes, higher VE was observed during the entire follow-up period. Among individuals above age 65, the mRNA vaccines showed better VE against infection than AZD1222 but similar high VE against hospitalization. Conclusions: Our findings provide strong evidence for long-term maintained protection against severe COVID-19 by the basic two-dose schedule, supporting more efforts to encourage unvaccinated persons to get the basic two doses, and encourage vaccinated persons to get a booster to ensure better population-level protection.

## 1. Introduction

With the rapid evolution of SARS-CoV-2 and vaccine approvals [1,2], concerns remain about long-term vaccine effectiveness (VE) against new variants and for newly approved vaccines. A meta-analysis of 18 studies until November 2021 reported waning VE against COVID-19 infection from 83% in the first month to 22% at five months or longer. Effectiveness against hospitalization or more severe outcomes was higher [3]. However, the meta-analysis did not include the period with Omicron, which was first detected in November 2021 and quickly became the dominant variant globally [4,5]. A rapid increase in COVID-19 infections, even in vaccinated populations, was seen in many countries and triggered concerns about the effectiveness of approved vaccines against Omicron. Early laboratory data also reported lower antibody response to Omicron than other strains of SARS-CoV-2 [6,7]. Several early studies with real-world settings further revealed lower VE and faster waning against Omicron infection in the UK, Qatar, and Malaysia [8,9,10].

In Sweden, vaccination was initiated in the elderly population on 27 December 2020 and reached larger and younger populations in 2021 and 2022 [11]. We used comprehensive Swedish register data to estimate the time-varying VE in reducing the risks of COVID-19 infection, hospitalization, intensive care unit (ICU) admission, and death in a 13-month follow-up and compared the pattern of time-varying VE before and after the emergence of Omicron.

## 2. Material and Methods

### 2.1. Study Design and Population

This study is part of the RECOVAC (Register-based large-scale national population study to monitor COVID-19 vaccination effectiveness and safety) study within the larger SCIFI-PEARL (Swedish COVID-19 Investigation for Future Insights—a Population Epidemiology Approach using Register Linkage) project with regularly updated data from various National Registers [12]. The current study included the whole Swedish population ≥ 12 years old in 2021, representing the approved population for COVID-19 vaccination in Sweden. We followed the cohort from 1 January 2020 (before the start of the pandemic) to 31 January 2022, with vaccines being introduced during the follow-up (first vaccination on 27 December 2020). The end of follow-up coincides with the termination of large-scale COVID-19 polymerase chain reactions (PCR) testing in Sweden. For COVID-19 ICU admission, the end of follow-up was 31 December 2021 due to data availability. The first Omicron case was diagnosed on 29 November 2021 in Sweden and quickly became the dominating variant (Appendix A) [13]. Consequently, we also subdivided the follow-up into before and after 1 December 2021, representing the emergence of Omicron. This study focused on two doses of vaccine, which was the originally recommended basic COVID-19 vaccination strategy.

### 2.2. Data Sources

We obtained data from multiple National Registers. Vaccination data came from the National Vaccination Register (NVR). All individuals with their first positive SARS-CoV-2 PCR test were identified from SmiNet, the national register of notifiable communicable diseases. PCR testing was introduced in 2020, and large-scale testing started in mid-2020. All individuals with symptoms of COVID-19 were encouraged to get tested, free of cost, until February 2022. COVID-19 diagnoses from outpatient specialist visits and inpatient care records were obtained from the Swedish National Patient Registry (NPR). COVID-19-related ICU data came from the Swedish Intensive care Register (SIR). The date and cause of death were obtained from the Register of Total Population (RTB) and the National Cause-of-Death Register (NCDR).

A complete medical history from 2015 was obtained from NPR and drug history for prescription drugs from 2018 from the National Prescribed Drug Register (NPDR). Sociodemographic data, including education, family situation, income, and occupation from 2015, were obtained from Statistics Sweden (SCB). Information on elderly subjects living at special care facilities and/or receiving home care services was obtained from the National Social Service Register.

## 3. Exposure and Outcomes

The exposure variables were vaccination status (unvaccinated, dose one, dose two), time intervals after each vaccination, and different vaccines (BNT162b2, mRNA-1273, and AZD1222), based on data from NVR. The first dose was defined as each individual’s first record in NVR. The second and third doses were defined as the following records with a predefined minimum time gap between doses (details see Appendix A).

Four different COVID-19 outcomes were investigated: COVID-19 infection; hospitalization; ICU admission; and death. COVID-19 infection was defined as the first of either a positive PCR test from SmiNet, a COVID-19 diagnosis code (ICD10: U07.1/U07.2) from NPR, an ICU admission from SIR, or death due to COVID-19 (underlying or contributing cause of death) from NCDR. Most COVID-19 infection cases (98.4%) were defined by positive PCR tests. The onset date of infection was set to two days before the registered date of any component event based on an estimated minimum incubation time [14]. For hospitalization and severe COVID-19 outcomes (ICU admission and death), the actual, registered date was used as the event date.

## 4. Covariates

The procedure of covariate selection was performed in 10% random samples of the data due to computational challenges related to the large population and dataset (Appendix A, Appendix A). We included the following covariates in the final models: age (modeled by restricted cubic spline with four knots), sex, country of birth (Sweden/other countries), health care workers (yes/no), income (tertiles of the study populations), education (primary, secondary, tertiary, unknown), marital status (married, unmarried, unknown), living at special housing and/or receiving home services for the elderly (yes/no), and prior comorbidities and treatments (yes/no). Prior comorbidities, including cardiovascular diseases, stroke, hypertension, diabetes, obstructive respiratory diseases, chronic kidney diseases, obesity, autoimmune diseases, dementia, psychiatric conditions, and cancer, were defined based on five-year prior medical history from NPR and prior treatments based on one-year prior prescription drug history from NPDR. Other covariates were defined with information retrieved from Statistics Sweden (Appendix A).

## 5. Statistical Analysis

We studied the first occurrence of each outcome during the pandemic; therefore, the VE estimates apply to the first occurrence of an outcome event after vaccination, compared to unvaccinated individuals, in individuals previously free of this event.

Cox proportional hazard models with time-varying exposure were used [15]. In the model, each individual’s follow-up time was first divided according to vaccination status (unvaccinated, first dose, and second dose), and then vaccination exposure periods were further divided into time intervals after each dose until the transition to the next dose (Appendix A). Since the fine division of follow-up time was computationally challenging, some modeling steps were performed in 10% random samples of the data to support the final full-scale analyses (Appendix A).

This paper focused on the VE for two doses of vaccine. In this analysis, the time period under the first dose was treated as a loss to follow-up, meaning subjects were observed during the unvaccinated period (since the study started to the time when receiving their first dose) and then again observed when receiving their second dose to the end of follow up, which occurred at an outcome event or censoring that is defined as third dose, emigration, death, or study ended, whichever came first. We also estimated the VE for one dose and reported results in the Appendix A.

To explore and illustrate VE trends in a smoother manner, we used an unrestricted cubic spline with five selected knots in extended Cox regression. We chose an unrestricted cubic spline for more flexibility in the two tails to adequately reflect the real-world data trends and ensure a better model fit. To cover the follow-up time since vaccination well, the five knots were selected at the 20th, 35th, 65th, 95th, and 99th percentiles.

We estimated time-varying VE for the entire follow-up, for the period before Omicron (until 30 November 2021) and for the Omicron period, respectively. For analysis of the Omicron period, we modeled the entire follow-up period, but only events after 1 December 2021 were considered as incident cases for estimation, and individuals with earlier events were censored at their event.

Additionally, stratified analyses were performed for COVID-19 infection and hospitalization by sex or age group (12–17, 18–39, 40–59, 60–64, 65–79, and 80+) and in subjects who received two doses of homologous BNT162b2, mRNA-1273, or AZD1222. As AZD1222 was primarily used in older individuals, stratified analyses were restricted to those aged 65 and older.

From the estimated hazard ratios (HR), results were presented as VE with 95% confidence intervals (CI), with VE calculated as 100 × (1 − HR). Chi-square tests were used to check the model fit, and all models showed *p* values < 0.0001. All analyses were performed in STATA MP 17 (StataCorp. 2021. Stata: Release 17).

## 6. Results

### 6.1. Study Population

Among 9,153,456 study individuals, 15% remained unvaccinated during the entire study period, 85% had at least one dose of vaccine, 82% had two doses, and 45% had ≥three doses as of 31 January 2022 (Table 1). Most individuals received two doses of BNT162b2 (78%) or mRNA-1273 (12%). Only 8% had two doses of AZD1222, and most (86%) were ≥65 years (Table 2). The average interval between the first and second and second and third doses was seven and 28 weeks, respectively. For homologous AZD1222, the interval between the first and second doses was slightly longer (ten weeks) and between the second and third doses slightly shorter (25 weeks) (Appendix A). Vaccine uptake trends are presented in Appendix A.

From 1 January 2020 to 31 January 2022, 2,002,024 first-time COVID-19 infections were reported (Table 1), representing 22% of the cohort. The corresponding figures for hospitalizations, ICU admissions, and deaths were 0.9%, 0.1%, and 0.2%, respectively.

### 6.2. VE during a 13-Month Follow-Up Period

The initial analysis was performed for the entire follow-up (13 months). After two doses of any vaccine, VE against COVID-19 infection peaked at week three at 72.0% (95%CI 71.0–73.0%) but then dropped quickly to 19.5% (18.8–20.2%) by weeks 14–17 and showed no protection from week 18 (Figure 1a, Appendix A). VE against hospitalization was above 82% from weeks one to 25 and peaked above 90% at weeks five to six (Figure 1b, Appendix A). VE after two doses against severe COVID-19 outcomes (ICU admission and death) was even higher and more durable (Figure 1c,d, Appendix A). Figure 2 shows unrestricted spline curves illustrating smoothed trends for all COVID-19 outcomes. The smoothed trends corresponded well with the time interval estimates shown in Figure 1. For the later period (e.g., after week 40), the curves are less stable as the last two knots are at the 95th and 99th percentiles in a time period with fewer data. Therefore, VE estimates from both time interval analysis and spline analysis during the later period should be interpreted with appropriate caution.

### 6.3. VE after Two Doses before and after the Emergence of Omicron

The more fast-waning VE observed for the entire follow-up than in other earlier published data appeared to be related to the emergence of Omicron. It is, therefore, important to analyze the pre-Omicron and Omicron periods separately. There was a large difference in VE against infection before and after the emergence of Omicron. VE was above 85% before Omicron in most time intervals (Figure 3a, Appendix A), whereas VE was lower and decreased rapidly during the Omicron period and two doses of the vaccine showed no protection against infection by week 14 (Figure 3b, Appendix A).

A difference in the VE between the pre-Omicron and Omicron period was also observed for hospitalization, but it was not as large as for infection (Figure 3c,d, Appendix A). Before Omicron, VE was stable, durable, and high (above 85%), while VE against hospitalization caused by Omicron was about 80% up to week 25 and then decreased but showed some protection against hospitalization during the entire follow-up.

### 6.4. VE after Two Doses by Age, Sex, and Vaccine Type

The overall VE trends against infection were relatively similar across age groups, although VE against infection was somewhat higher in older age groups (Appendix A). VE against hospitalization showed relatively few differences, with only a slightly higher VE being observed in the 40–60 and 60–65-year age groups (Appendix A). There was a slight sex difference (Appendix A), with lower VE in males. AZD1222 showed lower VE against infection than the mRNA vaccines (Appendix A). In terms of hospitalization, the three vaccines showed similar higher VEs in the early weeks, with AZD1222 showing an earlier decrease (Appendix A).

## 7. Discussion

This study examined the time-varying VE against infection, hospitalization, and severe COVID-19 outcomes over 13 months, including the emergence of the Omicron variant in Sweden. We modeled exposure over time with high granularity (first weekly, then monthly after vaccination) to provide a more detailed VE investigation than the previous vaccine studies. The most important finding was a difference in the pattern of VE during the pre-Omicron and Omicron periods. We found a high and stable VE (in the range of 85–95%) against infection when restricting the analysis to the pre-Omicron period, which was similar to a US study with over 10 million North Carolina residents and a nine-month follow-up until September 2021. They estimated monthly VE after two doses of the two mRNA vaccines, with a peak VE of about 95% at two months after the first dose, decreasing to 70–80% at seven months [16]. However, in a pre-Omicron Swedish study that included 1,685,948 individuals with 1:1 matching of vaccinated to unvaccinated, they found more progressively waning was observed with an estimated peak (92%) at 15–30 days, declining to no effect after eight months with two doses of vaccine [17]. The difference between our study and that study is likely due to the different target populations, as our study examined the whole population, while that study investigated only 30% of vaccinated individuals who could be matched [17].

We observed, however, very rapidly waning effectiveness during the Omicron period, which dropped to zero protection by week 14. Several studies have also reported lower and more rapid VE waning with Omicron [8,9,10,18,19]. All of these studies used a test-negative case-control study design with potential limitations such as being sensitive to the test sensitivity and specificity [20]. In a UK study [9], with two doses of mRNA vaccines, the VE dropped from 65–70% to 10% by 25 weeks and was even lower and less durable with two doses of AZD1222 (from 45–50% to no effect by 20 weeks). This is broadly consistent with our results.

Our data showed negative VE against Omicron infection from week 14, indicating that vaccinated individuals experienced a higher risk of infection than those unvaccinated. This is likely to be due to unbalanced susceptibility among vaccinated and unvaccinated individuals who had not developed COVID-19 by that time. In time-to-event analyses such as in this study, when an event is inevitable or very common, a protective effect in one group in the short term would be expected to reverse to increased risk at later time points as a reflection of the delay in disease onset in the protected group. The sudden emergence of the highly infective and less severe Omicron variant, with vaccine escape properties [21], represents a sudden change of the COVID-19 infection to a situation where it transitions from a moderately common infection to an almost ubiquitous infection that essentially everyone can expect to contract eventually, as reflected in the drastic VE reversal that we document in our study. Protection against severe disease, however, remains substantial.

Our analysis for the entire follow-up showed an overall lower peak VE than in previous studies, including phase-three trials [22,23,24,25,26,27] and observational studies [16,17]. This was due to the long follow-up period that covered the emergence of Omicron and the difference in VE during the pre-Omicron and Omicron periods. More importantly, greater and longer protection was seen against hospitalization and severe COVID-19 (ICU admission and death) than against COVID-19 infection in this study, as previously reported in other studies [16,17,28]. This study also confirms that VE against hospitalization remained high and durable even during the Omicron period [8,29].

In line with previous trials and real-world observational studies showing lower VE of AZD1222 than mRNA-1273 or BNT162b2 [17,28,30], we observed lower VE after two doses of homologous AZD1222 than the two mRNA vaccines among individuals ≥ 65 years. We restricted the age range in this analysis based on the Swedish vaccine strategy, where AZD1222 was offered to the older population and stopped in mid-2021 (Appendix A). For COVID-19 hospitalization, the three vaccines showed similar VE in the early period, with homologous AZD1222 waning faster from week 20. Overall, we saw similar satisfactory VE across sex and age groups, but we did observe a somewhat unexpected suggested higher VE for infection among older ages. This may be time-related since vaccinations were started with older age groups, and during that period, other social restrictions were stricter than later when vaccination was opened for younger ages. The early variants were also less easily infectious than Omicron. This observation emphasizes that real-life observed VE for being infected is a function of the biological vaccine effect and other aspects of infection spread.

The analysis in this study focused on two doses, as this was originally recommended basic vaccination schedule we wished to study. Despite the introduction of a third booster dose, some individuals and groups have considered themselves adequately covered by two doses, and the coverage of three vaccine doses has been relatively low to date. In addition, dose three had substantially fewer outcomes and a shorter follow-up period, which precludes a detailed time-related analysis such as the one we conducted here for two doses.

The population-based character of our study reduces the potential for selection bias. Additionally, we used Cox regression, considering both time-varying exposure (from unvaccinated to one dose and then two doses) and time-varying effects (period effects for each dose). This approach avoids difficult assumptions about the interval between doses, as in the mentioned US study [16]. However, our results can be influenced by human behaviors related to vaccination. Those who chose to be vaccinated later, or not at all, may behave differently from those who chose to be vaccinated earlier, a potential bias that is difficult to address. Additionally, as the number of home-based tests and antigen tests increases, there is a risk of missing COVID-19 infection cases. However, suboptimal sensitivity of outcome assessment is relevant for all observational studies, and the Swedish register data system nonetheless remains among the best in the world, capturing a broad range of outcomes, including COVID-19, with high accuracy.

## 8. Conclusions

This study provides more detailed long-term data on time-varying VE against COVID-19 in a complete general population. The progressively waning protection against Omicron infection after two vaccine doses underscores the need for additional efforts to encourage people to get a booster dose to ensure better population-level protection. With respect to hospitalization and severe COVID-19, although waning was stronger during the Omicron period than the pre-Omicron period, two doses of vaccine continued to provide very good and long-lasting protection, emphasizing the need to continue to encourage persons not yet vaccinated to receive their basic two-dose vaccination.

## Figures and Tables

**Figure 1 vaccines-10-02074-f001:**
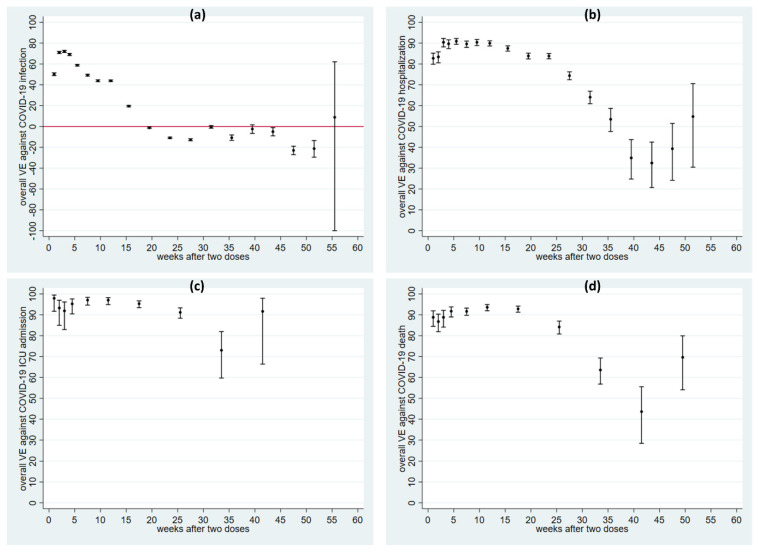
Adjusted vaccine effectiveness (dots) with 95% CI (capped spikes) against COVID-19 infection (panel (**a**)), hospitalization (**b**), severe outcomes [ICU admission (**c**), and death (**d**)] at each time interval after two doses. Note that the y-axis scales are different for infection (**a**) and the other outcomes. Legend: VE denotes vaccine effectiveness. The red line in (**a**) indicates VE = 0.

**Figure 2 vaccines-10-02074-f002:**
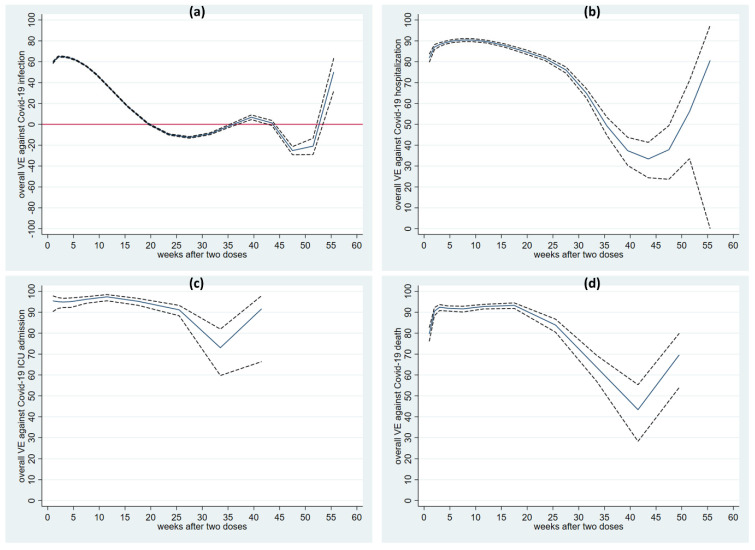
Unrestricted cubic splines of vaccine effectiveness against COVID-19 infection (panel (**a**)), hospitalization (**b**), severe outcomes [ICU admission (**c**), and death (**d**)] after two doses. Legend: VE denotes vaccine effectiveness. The area within the dotted line indicates 95% confidence intervals. Note that the y-axis scales are different for infection (**a**) and the others. The red line in (**a**) indicates VE = 0.

**Figure 3 vaccines-10-02074-f003:**
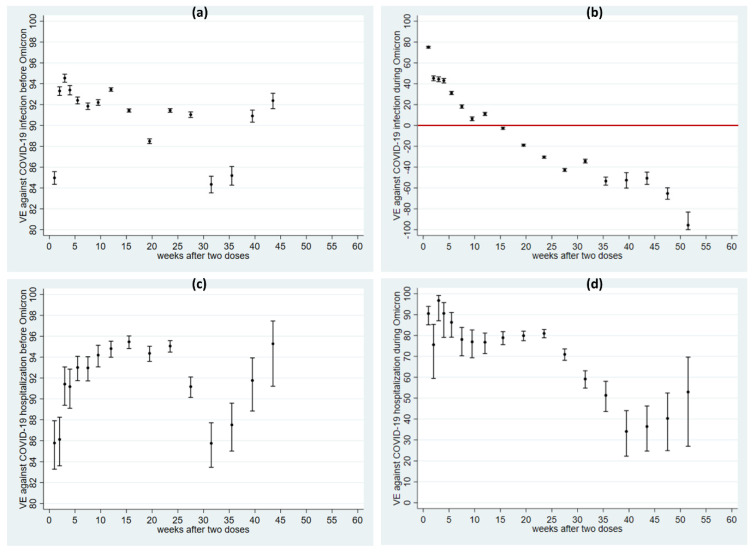
Adjusted vaccine effectiveness (dots) with 95% CI (capped spikes) of vaccine effectiveness before and after Omicron against COVID-19 infection (**a**,**b**) and COVID-19 hospitalization (**c**,**d**). Note that the y-axis scales are quite different. Legend: VE denotes vaccine effectiveness. The red line in (**b**) indicates VE = 0.

**Table 1 vaccines-10-02074-t001:** Sociodemographic and comorbidity characteristics of the study cohort according to vaccine uptake and COVID-19 outcomes by 31 January 2022.

	Entire Population	Vaccine Uptake						COVID-19 Outcome Events			
		Vaccinated with at Least One Dose		Vaccinated with at Least Two Doses		Vaccinated with More Than Two Doses		Infection	Hospitalization	ICU	Death
Characteristics	Count	Count	%	Count	%	Count	%	Count	Count	Count	Count
All residents ≥ 12 year	9,153,456	7,750,175	84.7	7,492,231	81.9	4,069,838	44.5	2,002,024	81,488	8167	17,408
**Age group**											
12–17 y	729,379	524,970	72.0	451,781	61.9	462	0.1	172,493	476	26	1
18–39 y	2,919,192	2,317,509	79.4	2,201,250	75.4	487,513	16.7	827,836	9388	601	76
40–59 y	2,636,496	2,304,475	87.4	2,262,200	85.8	1,289,901	48.9	719,142	20,626	2432	563
60–64 y	576,885	526,344	91.2	521,098	90.3	430,858	74.7	98,341	7226	1147	460
65–79 y	1,613,620	1,496,463	92.7	1,483,929	92.0	1,352,311	83.8	126,937	23,392	3355	4309
≥80 y	677,884	580,414	85.6	571,973	84.4	508,793	75.1	57,275	20,380	606	11,999
**Sex**											
Males	4,593,274	3,827,301	83.3	3,689,578	80.3	1,914,129	41.7	955,270	45,558	5704	9538
Females	4,560,182	3,922,874	86.0	3,802,653	83.4	2,155,709	47.3	1,046,754	35,930	2463	7870
**Country of birth**											
Sweden	7,173,976	6,375,825	88.9	6,201,225	86.4	3,557,726	49.6	1,561,900	53,760	4941	13,810
Other countries	1,979,480	1,374,350	69.4	1,291,006	65.2	512,112	25.9	440,124	27,728	3226	3598
**Health care workers**											
Yes	1,765,714	1,548,251	87.7	1,504,469	85.2	806,464	45.7	540,620	12,198	1304	414
No	7,387,742	6,201,924	83.9	5,987,762	81.0	3,263,374	44.2	1,461,404	69,290	6863	16,994
**Education**											
Primary	1,624,747	1,330,160	81.9	1,276,567	78.6	711,662	43.8	292,496	23,732	2288	7389
Secondary	3,515,542	3,037,726	86.4	2,961,176	84.2	1,729,812	49.2	771,477	33,412	3545	6450
Tertiary	3,033,764	2,728,483	89.9	2,684,092	88.5	1,586,958	52.3	731,879	21,109	2037	2978
Unknown	979,403	653,806	66.8	570,396	58.2	41,406	4.2	206,172	3235	297	591
**Income** ^(**a**)^											
Low	2,843,443	2,151,668	75.7	2,055,470	72.3	1,067,553	37.5	137,7828	101,658	7885	13,811
Medium	2,843,745	2,512,100	88.3	2,452,190	86.2	1,317,623	46.3	2,079,536	68,123	6560	5159
High	2,872,082	2,653,320	92.4	2,617,388	91.1	1,684,159	58.6	2,171,200	61,814	6675	2617
Unknown	624,186	433,087	69.4	367,183	58.8	503	0.1	330,659	794	51	3
**Marital status**											
Married	3,408,329	3,054,335	89.6	3,005,911	88.2	2,096,079	61.5	738,892	37,793	4355	6123
Unmarried	5,732,710	4,693,822	81.9	4,484,570	78.2	1,973,458	34.4	1,262,593	43,666	3810	11,284
Unknown	12,417	2018	16.3	1750	14.1	301	2.4	539	29	2	1
**Special care facilities**											
No	9,065,928	7,693,339	84.9	7,437,607	82.0	4,028,541	44.4	1,983,797	78,662	8135	11,832
Yes	87,528	56,836	64.9	54,624	62.4	41,297	47.2	18,227	2826	32	5576
**Home care service**											
No	8,903,449	7,559,894	84.9	7,307,304	82.1	3,922,014	44.1	1,964,729	68,803	7824	8818
Yes	250,007	190,281	76.1	184,927	74.0	147,824	59.1	37,295	12,685	343	8590
**Prior comorbidities and treatments** ^(**b**)^											
**Cardiovascular** **disease**											
No	8,478,793	7,162,241	84.5	6,914,482	81.6	3,594,586	42.4	1,911,159	61,569	6762	9537
Yes	674,663	587,934	87.1	577,749	85.6	475,252	70.4	90,865	19,919	1405	7871
**Stroke**											
No	9,058,839	7,670,273	84.7	7,413,794	81.8	4,004,535	44.2	1,989,901	78,182	7987	15,800
Yes	94,617	79,902	84.4	78,437	82.9	65,303	69.0	12,123	3306	180	1608
**Hypertension**											
No	7,113,536	5,892,738	82.8	5,657,425	79.5	2,513,924	35.3	1,743,515	38,670	3918	4184
Yes	2,039,920	1,857,437	91.1	1,834,806	89.9	1,555,914	76.3	258,509	42,818	4249	13,224
**Diabetes**											
No	8,610,137	7,266,301	84.4	7,015,964	81.5	3,684,709	42.8	1,923,342	64,901	6204	12,690
Yes	543,319	483,874	89.1	476,267	87.7	385,129	70.9	78,682	16,587	1963	4718
**Obstructive** **respiratory diseases**											
No	8,314,536	7,006,607	84.3	6,769,169	81.4	3,601,311	43.3	1,824,514	65,536	6648	13,652
Yes	838,920	743,568	88.6	723,062	86.2	468,527	55.8	177,510	15,952	1519	3756
**Chronic kidney** **diseases**											
No	9,055,450	7,671,976	84.7	7,415,874	81.9	4,008,312	44.3	1,987,063	75,642	7731	14,937
Yes	98,006	78,199	79.8	76,357	77.9	61,526	62.8	14,961	5846	436	2471
**Obesity**											
No	8,981,258	7,604,699	84.7	7,352,735	81.9	3,992,828	44.5	1,957,435	77,704	7678	16,835
Yes	172,198	145,476	84.5	139,496	81.0	77,010	44.7	44,589	3784	489	573
**Autoimmune** **diseases**											
No	8,943,975	7,564,864	84.6	7,310,314	81.7	3,927,986	43.9	1,967,287	75,473	7635	15,460
Yes	209,481	185,311	88.5	181,971	86.9	141,852	67.7	34,737	6015	532	1948
**Dementia**											
No	9,097,600	7,710,590	84.8	7,453,854	81.9	4,039,271	44.4	1,991,197	79,069	8149	14,309
Yes	55,856	39,585	70.9	38,377	68.7	30,567	54.7	10,827	2419	18	3099
**Psychiatric** **conditions**											
No	7,411,418	6,240,821	84.2	6,025,166	81.3	3,120,897	42.1	1,666,204	53,403	5898	7771
Yes	1,742,038	1,509,354	86.6	1,467,065	84.2	948,941	54.5	335,820	28,085	2269	9637
**Cancer**											
No	8,705,114	7,352,319	84.5	7,099,348	81.6	3,732,517	42.9	1,948,418	71,208	7429	13,814
Yes	448,342	397,856	88.7	392,883	87.6	337,321	75.2	53,606	10,280	738	3594

^(**a**)^ Low/medium/high income categorized using tertiles of the study populations. ^(**b**)^ Prior comorbidities and treatments were defined using information from the two-year prior medical history and one-year prior prescription drug history (Appendix A).

**Table 2 vaccines-10-02074-t002:** Sociodemographic and comorbidity characteristics of people receiving two doses according to vaccine type.

	Vaccine Uptake	Vaccine Type					
	Two Doses	Homologous BNT162b2		Homologous mRNA-1273		Homologous AZD1222	
Characteristics	Count	Count	%	Count	%	Count	%
All residents ≥ 12 year	7,492,231	5,858,168	78.2	894,487	11.9	595,039	7.9
**Age group**							
12–17 y	451,781	446,576	7.6	30,129	3.4	1	0.0
18–39 y	2,201,250	1,739,396	29.7	356,327	39.8	29,772	5.0
40–59 y	2,262,200	1,859,436	31.7	298,996	33.4	41,381	7.0
60–64 y	521,098	456,742	7.8	38,203	4.3	12,063	2.0
65–79 y	1,483,929	883,888	15.1	106,293	11.9	478,493	80.4
≥80 y	571,973	472,130	8.1	64,539	7.2	33,329	5.6
**Sex**							
Males	3,689,578	2,910,935	49.7	455,604	50.9	281,162	47.3
Females	3,802,653	2,947,233	50.3	438,883	49.1	313,877	52.7
**Country of birth**							
Sweden	6,201,225	4,827,821	82.4	724,205	81.0	524,892	88.2
Other countries	1,291,006	1,030,347	17.6	170,282	19.0	70,147	11.8
**Health care workers**							
Yes	1,504,469	1,112,637	19.0	179,500	20.1	119,297	20.0
No	5,987,762	4,745,531	81.0	714,987	79.9	475,742	80.0
**Education**							
Primary	1,276,567	971,760	16.6	162,404	18.2	121,539	20.4
Secondary	2,961,176	2,270,684	38.8	357,753	40.0	261,433	43.9
Tertiary	2,684,092	2,075,266	35.4	326,770	36.5	207,945	34.9
Unknown	570,396	540,458	9.2	47,560	5.3	4122	0.7
**Income** ^(**a**)^							
Low	2,055,470	1,583,045	27.0	266,250	29.8	166,084	27.9
Medium	2,452,190	1,865,049	31.8	313,698	35.1	203,912	34.3
High	2,617,388	2,026,669	34.6	300,081	33.5	225,007	37.8
Unknown	367,183	383,405	6.5	14,458	1.6	36	0.0
**Marital status**							
Married	3,005,911	2,276,394	38.9	330,761	37.0	339,868	57.1
Unmarried	4,484,570	3,580,270	61.1	563,482	63.0	255,135	42.9
Unknown	1750	1504	0.0	244	0.0	36	0.0
**Special care facilities**							
No	7,437,607	5,804,580	99.1	893,989	99.9	594,881	100.0
Yes	54,624	53,588	0.9	498	0.1	158	0.0
**Home care service**							
No	7,307,304	5,692,424	97.2	880,716	98.5	590,566	99.2
Yes	184,927	165,744	2.8	13,771	1.5	4473	0.8
**Prior comorbidities and treatments** ^(**b**)^							
**Cardiovascular disease**							
No	6914,482	5,428,674	92.7	837,515	93.6	510,530	85.8
Yes	577,749	429,494	7.3	56,972	6.4	84,509	14.2
**Stroke**							
No	7,413,794	5,797,064	99.0	887,348	99.2	585,524	98.4
Yes	78,437	61,104	1.0	7139	0.8	9515	1.6
**Hypertension**							
No	5,657,425	4,526,433	77.3	719,071	80.4	293,171	49.3
Yes	1,834,806	1,331,735	22.7	175,416	19.6	301,868	50.7
**Diabetes**							
No	7,015,964	5,509,183	94.0	846,257	94.6	5224,51	87.8
Yes	476,267	348,985	6.0	48,230	5.4	72,588	12.2
**Obstructive respiratory diseases**							
No	6,769,169	5,302,461	90.5	814,549	91.1	521,747	87.7
Yes	723,062	555,707	9.5	79,938	8.9	73,292	12.3
**Chronic kidney diseases**							
No	7,415,874	5,799,854	99.0	885,461	99.0	586,798	98.6
Yes	76,357	58,314	1.0	9026	1.0	8241	1.4
**Obesity**							
No	7352,,735	5,748,820	98.1	877,508	98.1	585,194	98.3
Yes	139,496	109,348	1.9	16,979	1.9	9845	1.7
**Autoimmune diseases**							
No	7,310,314	5,721,375	97.7	875,715	97.9	571,667	96.1
Yes	181,971	136,793	2.3	18,772	2.1	23,372	3.9
**Dementia**							
No	7,453,854	5,823,453	99.4	892,704	99.8	593,332	99.7
Yes	38,377	34,715	0.6	1783	0.2	1707	0.3
**Psychiatric conditions**							
No	6,025,166	4,729,802	80.7	727,596	81.3	460,745	77.4
Yes	1,467,065	1,128,366	19.3	166,891	18.7	134,294	22.6
**Cancer**							
No	7,099,348	5,576,635	95.2	856,102	95.7	527,112	88.6
Yes	392,883	281,533	4.8	38,385	4.3	67,927	11.4

^(**a**)^ Low/medium/high income categorized using tertiles of the study populations. ^(**b**)^ Prior comorbidities and treatments were defined using information from the two-year prior medical history and one-year prior prescription drug history (Appendix A).

## Data Availability

This study used pseudonymized individual-level data from Swedish healthcare registers that are not publicly available according to Swedish legislation. The data can be obtained from the respective Swedish data holders on the basis of ethics approval for the research in question, subject to relevant legislation, processes, and data protection.

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
