# Peer review of "Effectiveness of COVID-19 Vaccines over 13 Months Covering the Period of the Emergence of the Omicron Variant in the Swedish Population"

_vaccines, 2022, doi:10.3390/vaccines10122074_

Round 1

Reviewer 1 Report

In this study, Xu et al. used comprehensive Swedish register data to estimate the time-varying effectiveness of three COVID-19 vaccines against SARS-CoV-2 infection, hospitalization, ICU admission, and death over a time period that included the emergence of the Omicron variant. The sample size (study population=9,153,456 individuals) has sufficient power, and the experimental design was scientifically sound, with conclusion appropriately limited. Cox regression analysis was stratified by age, gender, and vaccine time. The authors found that the primary two-dose vaccine regimen provided sufficient protection against hospitalization and severe COVID-19, while waning protection was greater during the Omicron period, suggesting the necessity for booster vaccination.  Studies such as this one are important, given the prevalence of the Omicron variant.  My only comments concern the quality of presentation  Inconsistent font styles are used and should be corrected as per the journal's prescriptions.  Also, I suggest that the authors boldface the covariate categories in Tables 1 and 2 to make them more clear in the presentation of the data.  Otherwise, this is an excellent study that provides valuable insight into COVID-19 vaccine effectiveness.

Author Response

In this study, Xu et al. used comprehensive Swedish register data to estimate the time-varying effectiveness of three COVID-19 vaccines against SARS-CoV-2 infection, hospitalization, ICU admission, and death over a time period that included the emergence of the Omicron variant. The sample size (study population=9,153,456 individuals) has sufficient power, and the experimental design was scientifically sound, with conclusion appropriately limited. Cox regression analysis was stratified by age, gender, and vaccine time. The authors found that the primary two-dose vaccine regimen provided sufficient protection against hospitalization and severe COVID-19, while waning protection was greater during the Omicron period, suggesting the necessity for booster vaccination.  Studies such as this one are important, given the prevalence of the Omicron variant.  My only comments concern the quality of presentation  Inconsistent font styles are used and should be corrected as per the journal's prescriptions.  Also, I suggest that the authors boldface the covariate categories in Tables 1 and 2 to make them more clear in the presentation of the data.  Otherwise, this is an excellent study that provides valuable insight into COVID-19 vaccine effectiveness.

Response: We thank the reviewer for the positive feedback on the manuscript. We have checked the font style throughout the current version. We also changed the format for Tables 1 and 2 as suggested. Please note that as these changes did not change the content, therefore we did not mark them with check changes. 

Reviewer 2 Report

In general, this is a very important study, since it provides information on real life protection, instead of surrogate immune markers. Another strength is that it includes a very large study population, allowing for high powered statistical analyses. The overall quality of the study is high, the presentation is clear the conclusions are modest and well supported by the results. 

Specific comments:

Methods: lines 80-82 - this would be more appropriate for the discussion. 

Results/Discussion: can the authors comment on why the death rate was higher than the ICU admission rate?

Similarly, are there any speculations on why VE was higher in the elderly patient group (lines 235-6)? 

Author Response

Methods: lines 80-82 - this would be more appropriate for the discussion. 

Response: Thank you for the suggestion. We removed the sentence to the first paragraph of the discussion with slight rewording: “We modelled exposure over time with high granularity (first weekly, then monthly after vaccination) to provide more detailed VE investigation than the previous vaccine studies.” (lines 245-247 in the version with track changes).

Results/Discussion: can the authors comment on why the death rate was higher than the ICU admission rate?

Response: It is because some covid related death cases were not admitted to ICU. A lower ICU admission rate than death rate has been seen in different countries, especially during the early period of the pandemic. For instance, in France, the weekly new ICU admissions were about 3000 persons in Nov 2020, and the corresponding weekly deaths were about 4000 persons. Similarly, in the Netherlands, the weekly new ICU admissions were about 300 persons in Nov 2020 and the corresponding deaths were 500 (data comes from https://ourworldindata.org/). We did not include this in the discussion as it was not relevant to our study aim—the vaccine effectiveness against various covid outcomes.

Similarly, are there any speculations on why VE was higher in the elderly patient group (lines 235-6)? 

Response: Thank you for the comment. It is true that previous RCTs showed higher VE in the younger population than the older population, but our real-world observation data showed a slightly different pattern, although the difference between age groups was very small. One speculative reason is that it may be due to the different time periods for different age groups. The vaccine campaign was started with older age groups and during that period, other social restrictions (e.g., lockdown) were stricter than the period when the vaccination was open for younger ones. The early variants were also less easily infectious than omicron. Therefore, the older age group was better protected by these strict restrictions and showed a lower risk for covid infection (higher VE) in our study. If we look at covid hospitalization, this age difference was no longer observed.

We have added some sentences in the Discussion (lines 297-304 in the version with track changes): “Overall we saw similar satisfactory VE across sex and age groups, but we did observe a somewhat unexpected suggested higher VE for infection among older ages. This may be time-related, since vaccinations were started with older age groups, and during that period other social restrictions were stricter than later when vaccination was opened for younger ages. The early variants were also less easily infectious than omicron. This observation emphasizes that real-life observed VE for being infected is a function of biological vaccine effect and other aspects of infection spread”.

Reviewer 3 Report

This study examines the preventive effect of the COVID-19 vaccine using official data from the Swedish government. The study is commendable in that it verified the effectiveness of the vaccine before and after an outbreak of the Omicron strain and clearly showed that the preventive effect against the Omicron strain was weaker than the preventive effect against the older strain.

Note that information on the statistical evaluation of the relationship between the explanatory variables and the objective variables presented in Table 1 and Table 2 is not provided. We believe that Chi-squared test and residual analysis are necessary to ensure the accuracy of the results and discussion.

Author Response

Note that information on the statistical evaluation of the relationship between the explanatory variables and the objective variables presented in Table 1 and Table 2 is not provided. We believe that Chi-squared test and residual analysis are necessary to ensure the accuracy of the results and discussion.

Response: Thank you for the comment. Tables 1 and 2 are purely descriptive to support the interpretation of the Cox regression results for the various groups, not for statistically comparing the groups, and thus statistical tests would not be appropriate in this setting in our view. The Chi-square test for all our Cox models showed p values<0.001, which is obvious as most of the VEs (converted from HRs from the model) were significant with confidence intervals away from the null. We have added one sentence in the Method (lines 164-165 in the version with track changes).

The residual analysis is one way to test the proportional hazards assumption. However, testing for proportional hazards has been recommended against and it has been pointed out that it is not a requirement for Cox regression as such to be valid (Stensrud & Hernán, 2020), since hazards can very rarely be assumed to be truly completely proportional, and the significance of the test will thus largely be driven by sample size. A more proper approach is to interpret the HR correctly, i.e., a summary HR is a weighted average of the true HRs over the entire follow-up period. Additionally, to get a better HR estimation closer to the true HR at various times, if it does vary over time, one can divide the follow-up into several time periods (Scenario 1 as discussed in that paper). In our study, we used such an approach and divided the follow-up time into smaller intervals (first weekly, then monthly), which provided HR for each interval which is then finally converted to time-varying VEs as reported in the paper. The interpretation of the reported V fairly well reflects the effects at time points along the follow-up. Based on these considerations we decided not to provide any residual analysis or other PH assumption test.

Ref: Stensrud, M. J., & Hernán, M. A. (2020). Why Test for Proportional Hazards? JAMA, 323(14), 1401.https://doi.org/10.1001/jama.2020.1267